# SFT Without Overfitting: Analyzing the Training Dynamics of Supervised Fine-tuning

## Abstract

Despite its central role in the post-training of large language models (LLMs), supervised fine-tuning (SFT) is prone to memorization and often fails to generalize to out-of-distribution (OOD) inputs. In this work, we present an empirical study of how different Transformer modules contribute to OOD generalization in rule-based reasoning tasks. We investigate the effect of selective fine-tuning, where the parameters of either feedforward neural networks or attention layers are updated during training. Our results show that fine-tuning only the attention layers substantially improves OOD generalization, while full-parameter or feedforward neural networks only tuning predominantly increases memorization and leads to generalization collapse. Remarkably, attention-only SFT achieves performance comparable to state-of-the-art reinforcement learning (RL) alignment methods. These findings provide new insights into the mechanisms underlying SFT and highlight selective SFT as a promising direction for improving the SFT generalization. We will release the code upon paper acceptance.

## 1 Introduction

Supervised fine-tuning (SFT) is a crucial step in adapting large language models (LLMs), ensuring stable and well-structured outputs for downstream tasks Chu et al. (2025); Guo et al. (2025). However, SFT has been shown to cause excessive memorization of training data, limiting generalization to out-of-distribution (OOD) inputs Chu et al. (2025); Guo et al. (2025), whereas subsequent reinforcement learning (RL) fine-tuning achieves much better cross-domain generalization.

Notably, the internal effects of SFT and RL on model parameters are quite different. Mukherjee et al. (2025) find that RL fine-tuning (including popular methods like PPO, DPO, GRPO etc.) updates only a small subnetwork, roughly 5–30% of the model's weights, leaving the majority of parameters essentially unchanged. In contrast, standard SFT induces far more widespread weight updates, as evidenced by much denser gradient changes during the SFT stage compared to the sparsity of RL-induced updates. This disparity may help explain why SFT is prone to overfitting: it alters a larger portion of the model's knowledge, whereas RL's more localized updates tend to preserve generalizable pre-trained knowledge. Indeed, RL-induced models often retain broader capabilities, whereas SFT models can become narrowly specialized to the fine-tuning data.

Our work is inspired by these differences and asks further: which modules of the Transformer architecture are most responsible for memorization and, consequently, for out-of-distribution generalization during SFT? In this work, we study out-of-distribution generalization in supervised fine-tuning, with a particular focus on the role of different Transformer modules.

Our primary contributions are as follows:

> **Contribution**
>
> 1. We are the first work to study out-of-distribution generalization in reasoning tasks under selective supervised fine-tuning.
> 2. Through controlled experiments, we analyze how memorization dynamics depend on different Transformer modules. We show that feedforward neural networks are prone to memorization and fail OOD generalization, while attention-only fine-tuning significantly improves generalization.
> 3. Extending beyond prior work (Chu et al., 2025) which reported that SFT uniformly fails to generalize OOD scenarios, we demonstrate that smaller learning rates prevent collapse, allowing SFT to achieve OOD performance competitive with, and in some cases surpassing, RL-based methods.

Together, these findings provide new insights into the training dynamics of SFT, highlighting how memorization emerges across Transformer modules and offering practical strategies for improving OOD generalization.

## 2 RELATED WORK

### 2.1 OUT-OF-DISTRIBUTION GENERALIZATION

Supervised fine-tuning has become the standard first step in adapting pretrained LLMs to alignment and reasoning tasks. However, it is increasingly recognized that SFT tends to overfit to training data and leads to poor generalization on out-of-distribution (OOD) inputs. Chu et al. (2025) shows that SFT memorizes task-specific patterns while reinforcement learning (RL) generalize in rule based reasoning tasks. The tension here is critical: SFT is indispensable for stabilizing model outputs, yet its propensity for memorization undermines cross-task adaptability. Prior work such as Mireshghallah et al. (2022) has mostly studied memorization in terms of privacy leakage, but to our best knowledge, no studies relate to how memorization affects OOD reasoning.

### 2.2 ROLES OF TRANSFORMER MODULES IN KNOWLEDGE AND MEMORY

Transformer architectures consist of attention layers and feedforward neural networks (FNNs), which serve different functions. Attention heads are generally associated with dynamic, context-sensitive processing and in-context learning, whereas FNNs are hypothesized to act as long-term memory stores. Geva et al. (2021) demonstrates that FNN layers behave like key-value memories, encoding textual associations within their weight matrices. Similarly, Meng et al. (2022) show that factual knowledge can be directly edited by modifying weights in mid-layer FNNs. Mireshghallah et al. (2022) empirically shows that fine-tuning layers closer to the logit output layer causes more likely training data leakage.

Although these works identify knowledge storage mechanisms, they do not directly study Transformer module-level contribution to OOD generalization.

**Positioning of Our Work** This paper addresses the gap by investigating the module-level contributions of memorization during SFT. We hypothesize that feedforward layers, as the main long-term memory stores, drive overfitting to fine-tuning data, reducing OOD reasoning performance. In contrast, updating attention layers may support adaptation with less risk of overwriting generalizable knowledge. By selectively freezing and fine-tuning different modules, we directly evaluate their impact on memorization and OOD reasoning generalization. This perspective reframes selective SFT not merely as a computationally efficient strategy, but as a targeted intervention for improving out-of-distribution reasoning.

## 3 METHODOLOGY

We investigate the contribution of distinct Transformer modules to the OOD generalization of supervised fine-tuning. Our approach is empirical: we selectively fine-tune target modules of a pre-trained model on instruction-formatted downstream reasoning datasets. Experiments are conducted on two

controlled environments: 1) GeneralPoints, an arithmetic reasoning benchmark, and 2) V-IRL, a language navigation task, both of which provide systematic evaluation to evaluate reasoning and generalization Zhai et al. (2024). Please see section 4.1 for the datasets detail.

Following Chu et al. (2025), we assess whether models trained with SFT acquire rule-generalizable knowledge or primarily memorize training-specific patterns. For each task, the model is fine-tuned on a single rule and subsequently evaluated both on the trained rule (in-distribution) and on previously unseen rule variants (out-of-distribution).

## 3.1 Supervised Fine-Tuning

Supervised fine-tuning adapts a pretrained language model to human-provided demonstrations by minimizing the discrepancy of reference responses. The mathematical definition of SFT is below:

We are given:

- A pretrained model $\pi_{\theta_0}(y \mid x)$, parameterized by $\theta_0$, where $x \in \mathcal{X}$ is an input (e.g., prompt) and $y \in \mathcal{Y}$ is an output (e.g., response).
- A supervised dataset
$$\mathcal{D} = \{(x_i, y_i^*)\}_{i=1}^N,$$
where each $x_i$ is a prompt and $y_i^*$ is the ground-truth or high-quality response.

The SFT objective is to minimize the cross-entropy loss of the labeled responses under the fine-tuned policy:

$$\mathcal{L}_{\text{SFT}}(\theta) = -\mathbb{E}_{(x,y^*)\sim\mathcal{D}} \left[\log \pi_\theta(y^* \mid x)\right] \tag{1}$$

**Token-level formulation:** If each response $y^* = (a_1, \ldots, a_T)$ is a sequence of $T$ tokens, then

$$\mathcal{L}_{\text{SFT}}(\theta) = -\mathbb{E}_{(x,y^*)\sim\mathcal{D}} \left[\sum_{t=1}^{T} \log \pi_\theta\big(a_t \mid x, y_{<t}^*\big)\right] \tag{2}$$

## 3.2 Transformer Modules

To isolate the contribution of different Transformer modules, we adopt a selective fine-tuning paradigm in which one of the following is updated during SFT:

- **FNN-only:** fine-tuning feedforward neural networks,
- **Attention-only:** fine-tuning self-attention layers,
- **Full-model:** fine-tuning all parameters.

Table 1: Experimental settings for selective fine-tuning.

| Setting | Trainable Parameters | Iterations |
|---------|---------------------|------------|
| **#Parameter not matched** | Different across modules | Different (fewer params → more steps) |
| **#Parameter matched** | Same across modules | Same |

**For a fair comparison**, we set the same compute budget (total training FLOPs), but the number of parameters in modules may vary in order to compare the default unmatched setting and the fair matched setting. Table 1 summarizes two experimental setups. Modules with fewer trainable parameters (e.g., attention-only) are trained for more iterations. This setup allows us to evaluate how each module affects SFT outcomes **without being biased by the module's capacity or training budget.**

## 4 Experimental Setup

Our experiments closely follow Chu et al. (2025).

## 4.1 DATASETS AND OOD REASONING TASKS

We evaluate memorization and generalization using two controlled reasoning tasks:

**GeneralPoints (GP):**

- Adapted from Zhai et al. (2024); Chu et al. (2025).
- Arithmetic card game where the model must form an expression equaling a target number (24 by default) using four given cards exactly once.
- Cards are described in natural language (e.g., "a red queen" → Q).
- OOD settings: enforced by varying symbolic mappings (e.g., treating face cards as 10 vs. 11, 12, 13).

**Virtual Intelligence in Real Life (V-IRL):**

- Adapted from Yang et al. (2024); Chu et al. (2025).
- Large-scale language navigation environment.
- Input: natural language route instructions paired with street-level imagery (textualized).
- Output: navigation actions to reach the correct destination.
- OOD settings: tested by altering action specifications (absolute vs. relative orientation) Chu et al. (2025).

These two benchmarks are chosen because they allow **precise control over symbolic rules**, making them well-suited for disentangling memorization from reasoning generalization. In **GP**, altering the numerical interpretation of face cards creates a controlled shift in symbolic rules, directly testing whether models adapt reasoning strategies or merely memorize mappings. In **V-IRL**, changing from absolute to relative action specifications probes the model's ability to transfer navigation policies across rule systems. Together, these benchmarks provide complementary views of rule-based reasoning: GP stresses symbolic arithmetic composition, while V-IRL evaluates instruction-following and spatial reasoning.

## 4.2 EVALUATION METRICS

For GeneralPoints, performance is measured by the success rate of producing a valid expression equaling the target Chu et al. (2025). For V-IRL, we report per-step accuracy (local action correctness relative to expert demonstrations) Chu et al. (2025). For both benchmarks, we distinguish between in-distribution (training rule) and out-of-distribution (novel rules) performance, thereby isolating memorization from generalization.

## 4.3 MODEL ARCHITECTURE AND TRAINING

Following Chu et al. (2025), we adopt Llama-3.2-Vision-11B Dubey et al. (2024) as the backbone model. SFT is conducted on expert demonstrations formatted as prompt–response pairs. All experiments were performed on $8 \times$ A100 GPUs (80 GB) with matched compute budgets across conditions.

## 5 RESULTS AND DISCUSSION

We now present our empirical findings on how selective fine-tuning on Transformer modules affects memorization and out-of-distribution generalization.

### 5.1 EFFECT OF MODULE CHOICE ON MEMORIZATION AND GENERALIZATION

Figure 1 shows the in-distribution (left) and out-of-distribution (right) performance of different fine-tuning strategies as a function of floating-point operations (FLOPs), across two reasoning benchmarks: GP (top) and V-IRL (bottom). We compare full fine-tuning, selective fine-tuning of feed-forward networks (labeled as FNN), and selective fine-tuning of attention layers (labeled as Attn).

To ensure a fair comparison, configurations with fewer trainable parameters (e.g., attention-only fine-tuning) are trained for proportionally more iterations, such that the total training FLOPs are approximately matched to those of full fine-tuning.

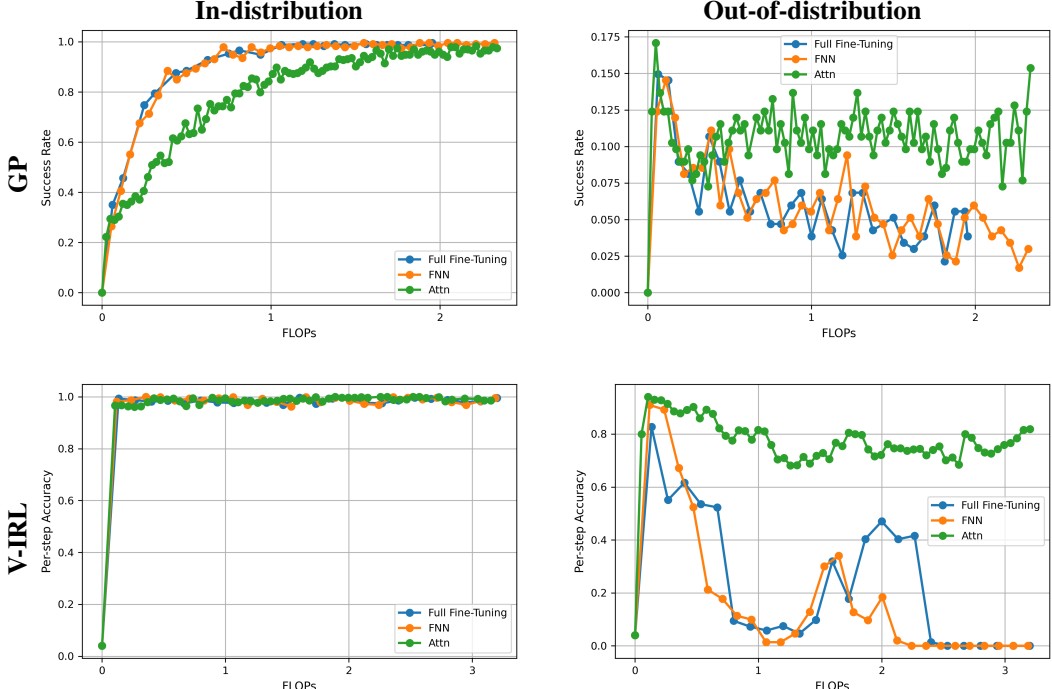

Figure 1: Performance vs. FLOPs ($10^{19}$) on GP and V-IRL for SFT with selected fine-tuned modules. FNN-only finetuning consistently exhibits performance degradation across all the OOD evaluations on all tasks, whereas attention-only fine-tuning preserves the OOD generalization. SFT with fewer trainable parameters (e.g., attention-only fine-tuning) are trained for additional iterations to match the total training FLOPs to full fine-tuning.

On in-distribution data, all three strategies quickly achieve near-perfect performance as FLOPs increase. This suggests that either module type can be tuned to fit the training distribution, and memorization allows the model to saturate performance regardless of which modules are updated. Notably, full fine-tuning and FNN-only tuning converge slightly faster on GP benchmark, consistent with FNNs' role as long-term memory stores that can readily encode training examples.

The out-of-distribution results, however, reveal a stark contrast. Both full fine-tuning and FNN-only tuning suffer degradation: success rates on GP collapse dramatically, and per-step accuracy on V-IRL drops to almost zero as training progresses. This highlights a classic memorization effect: these strategies overfit to training patterns and catastrophically fail to generalize to unseen scenarios. By contrast, attention-only fine-tuning sustains substantially higher OOD performance: on GP benchmark, attention-only fine-tuning maintains success rates above 10%, outperforming the other modules fine-tuning by a wide margin. On V-IRL benchmark, attention-only fine-tuning preserves per-step accuracy above 70–80% even at later stages, whereas other strategies degrade almost entirely.

> **Takeaway 1: Attention-only fine-tuning prevents OOD collapse**
>
> Fine-tuning feedforward layers or the full model achieves strong in-distribution performance but collapses on out-of-distribution tasks, while attention-only fine-tuning consistently preserves out-of-distribution generalization.

**SFT Under Matched Numbers of Trainable Parameter** In addition, we conduct experiments under a parameter-matched setting, where the number of trainable parameters is kept nearly identical

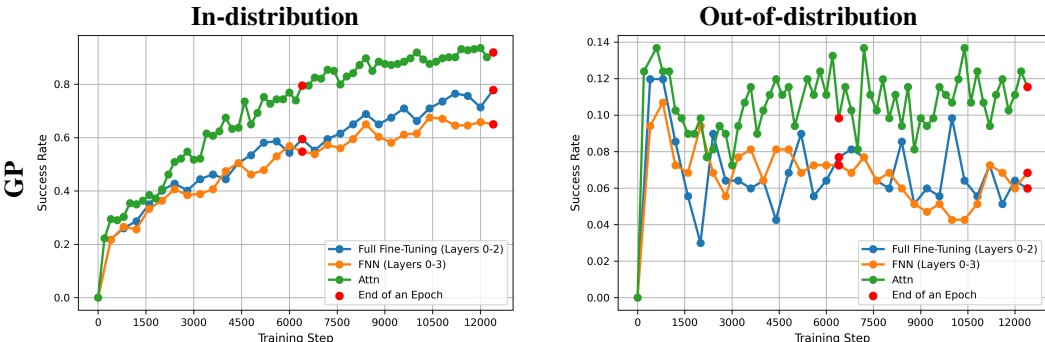

Figure 2: Performance vs. Training steps on GeneralPoints and V-IRL for SFT with selected fine-tuned modules and *matched number of trainable parameters*.

across configurations. Specifically, we restrict fine-tuning to the first $L$ transformer blocks rather than the entire model, with $L$ chosen to balance parameter counts. Let $N(\theta_{\text{attn}})$ denote the number of trainable parameters in all attention layers, and $N(\theta_{0:L}^{\mathcal{M}})$ denote the cumulative number of trainable parameters in module $\mathcal{M}$ (either FNNs or transformer blocks) from layer 0 up to $L$. We then select $L$ as:

$$L = \arg\min_{l} \|N(\theta_{\text{attn}}) - N(\theta_{0:l}^{\mathcal{M}})\|_2^2 \tag{3}$$

Figure 2 shows the in-distribution (left) and out-of-distribution (right) performance on GP benchmark under parameter-matched setting.

On the in-distribution benchmark, all methods improve steadily with training, though important differences emerge. Full fine-tuning and FNN-only tuning achieve moderate success rates (70–75%) after convergence, but attention-only fine-tuning substantially outperforms both, reaching close to 100% success rate. This indicates that even under identical parameter, and correspondingly compute budgets, attention layers provide a more efficient fine-tuning for fitting the training distribution.

The divergence is more pronounced in the out-of-distribution evaluation. Full fine-tuning and FNN-only tuning plateau at low success rates (6–8%) and exhibit strong instability, reflecting severe overfitting to the training distribution. In contrast, attention-only fine-tuning maintains consistently higher performance, with success rates in the 10–15% range throughout training. The robustness of attention tuning under matched training FLOPs highlights that its generalization benefits are not simply a byproduct of fewer parameters being updated, but instead stem from the functional role of attention modules in supporting flexible, context-sensitive adaptation.

These results in Figure 1 and Figure 2 strengthen our central claim: memorization during SFT is tied to FNN updates, while restricting updates to attention layers preserves generalizable knowledge and leads to superior OOD reasoning. Importantly, this advantage persists even when accounting for training efficiency, showing that selective attention tuning is a principled strategy for improving reasoning generalization without additional computational overhead.

> **Takeaway 2: Attention wins under matched number of trainable parameters**
>
> Even under matched numbers of trainable parameters, attention-only tuning yields superior in-distribution performance and significantly better OOD generalization, confirming attention layers' generalizability beyond computational fairness concerns.

### 5.2 Impact of Learning Rate on Memorization vs. OOD Generalization

Figure 3 illustrates the effect of learning rate on supervised fine-tuning memorization, comparing three settings ($1e$-6, $1e$-7, $1e$-8) across both in-distribution (left) and out-of-distribution (right) evaluations on GP (top) and V-IRL (bottom). For all experiments in this section, we report SFT results under full fine-tuning.

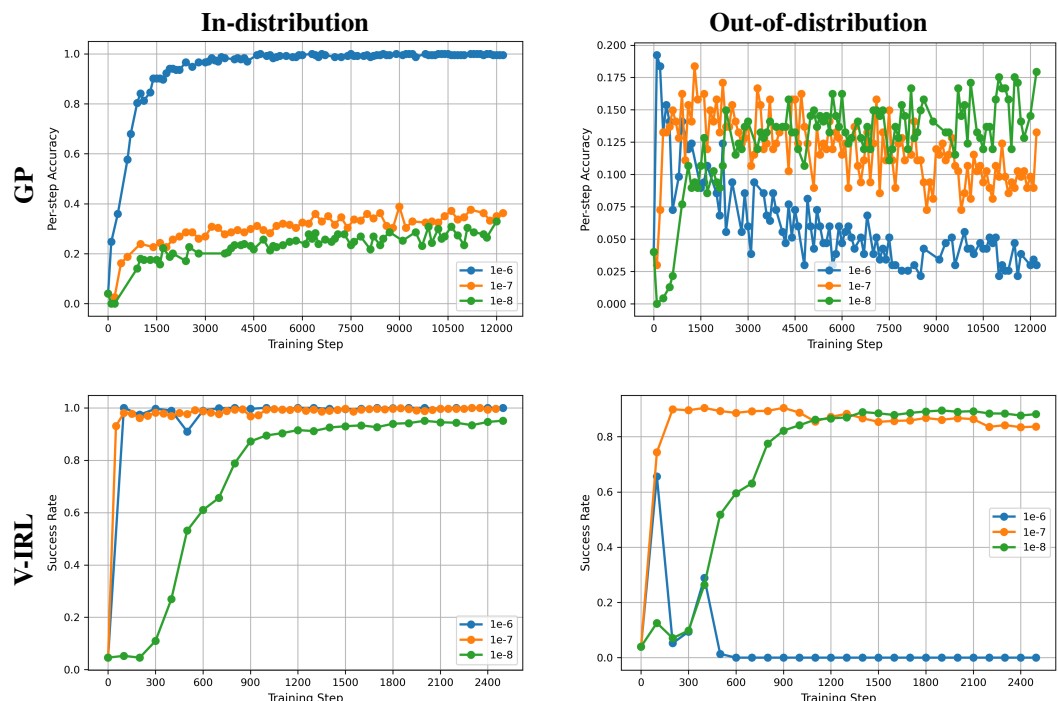

Figure 3: Effect of Learning Rate on SFT Memorization on V-IRL and GP benchmarks.

In contrast to Chu et al. (2025), our experiments demonstrate that substantially smaller learning rates can alter the OOD trend (see figure 3), allowing SFT to match or even surpass reinforcement learning in OOD settings (see table 2). On in-distribution tasks, a higher learning rate (1$e$-6) leads to rapid convergence. In GP, the model with 1$e$-6 achieves near-perfect per-step accuracy within 2,000 steps, whereas smaller learning rates converge more slowly and to lower ceilings. Similarly, in V-IRL, 1$e$-6 and 1$e$-7 quickly achieve near 100% success rates, while 1$e$-8 lags significantly before eventually catching up. This suggests that high learning rates accelerate memorization of training data.

On out-of-distribution tasks, however, the pattern reverses. Models fine-tuned with 1$e$-6 suffer from severe generalization collapse. In GP, OOD accuracy for 1$e$-6 drops below 5% after a few thousand steps, while 1$e$-7 and 1$e$-8 maintain substantially higher and more stable accuracy (10–17%). A similar trend appears in V-IRL: 1$e$-6 initially peaks but rapidly degrades to near-zero success rates, whereas lower learning rates (1$e$-7, 1$e$-8) stabilize around 70–80% success rates.

These results provide strong evidence that large learning rates amplify memorization and harm OOD generalization, while smaller learning rates act as a regularizer, slowing memorization and preserving the model's reasoning capabilities on unseen distributions. Importantly, this demonstrates that beyond which modules are updated (Figures 1-2), the magnitude of parameter updates is another key factor in controlling memorization during SFT.

| Method | V-IRL | GP |
|---|---|---|
| SFT (FFT) + RL Chu et al. (2025) | 91.8 | 15.0 |
| SFT (Full Fine-Tuning) | 89.05 | 17.95 |
| SFT (FNN-only) | 91.28 | 16.79 |
| SFT (Attention-only) | **94.13** | **19.23** |

Table 2: Effectiveness of LLM modules in SFT out-of-distribution generalization. Success rate (GP) and per-step accuracy (V-IRL) are used as evaluation metrics. Attention-only fine-tuning significantly improves generalization, performing on par with or better than state-of-the-art SFT+RL methods.

> **Takeaway 3: Small Rates, Big Gains**
>
> Building on prior work Chu et al. (2025) that found SFT fails to generalize OOD, we show that using smaller learning rates prevents collapse and enables SFT to reach, or even exceed, RL-induced OOD performance.

## 6 CONCLUSION

We conducted a systematic study of supervised fine-tuning in large language models, focusing on how different Transformer modules affect out-of-distribution generalization. Our experiments show that fine-tuning only attention layers preserves OOD performance, while FNN-only or full-model fine-tuning leads to memorization and fails generalization. Importantly, this advantage holds even under matched trainable parameters and total training FLOPs. In addition, we further showed that learning rate plays a critical role: smaller rates mitigate collapse and unlock stronger OOD performance.

These findings highlight that selective attention-only SFT is a simple, computationally efficient strategy to mitigate memorization and enhance OOD reasoning generalization. Our work provides practical guidelines for improving SFT and offers new insights into the module-level dynamics of memorization in large language models.

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
