# A  APPENDIX

## A.1  FLOPs CALCULATION

**Full Fine Tuning and Inference FLOPs**  We estimate the training FLOPs for Full Fine Tuning and inference following Kaplan et al. (2020) as:

$$Full\ Finetuning\ Train\ Flops = 6pn_{train} \tag{4}$$
$$Inference\ Flops = 2pn_{inf}, \tag{5}$$

where $p$ represents the model parameters, and $n_{train}$ and $n_{inf}$ represents the number of train and inference tokens, respectively.

**Selective Fine Tuning FLOPs**  Since the computational cost of the backward pass is approximately twice as the forward pass, following Zhou et al. (2024) we modify the formula for as:

$$Selective\ Finetuning\ Train\ Flops = (2p_f + 4p_t) \times n_{train}, \tag{6}$$

where $p_f$ and $p_t$ represent the number of frozen and trainable parameters respectively.

## A.2  DECLARATION OF USING GENERATIVE AI

Generative AI was used only for polishing the writing and improving readability. It was not involved in generating research ideas, experiments, or results.