# OpenReview forum: "SFT WITHOUT OVERFITTING: ANALYZING THE TRAINING DYNAMICS OF SUPERVISED FINE-TUNING"
_ICLR.cc/2026/Conference — ICLR 2026 Conference Withdrawn Submission_

### Official Review · Reviewer_ASpB · 2025-10-19

**Soundness:** 2
**Presentation:** 1
**Contribution:** 1
**Rating:** 2
**Confidence:** 4

**Summary:**

Through controlled experiments on reasoning tasks, the study empirically demonstrates that which Transformer modules are updated during SFT critically impacts this trade-off between memorization and generalization.

**Strengths:**

> s1: Selective freezing and fine-tuning of specific modules presents a reasonable starting point.


> s2: In analyzing generalization performance, it evaluates both in-distribution (ID) and out-of-distribution (OOD) scenarios.

**Weaknesses:**

>w1: **Insufficient literature review**: Lines 80-81: "to our best knowledge, no studies relate to how memorization affects OOD reasoning". However, several relevant works should be discussed [1,2].
>
>[1] Task Generalization with Autoregressive Compositional Structure: Can Learning from $D$ Tasks Generalize to $D^T$ Tasks? ICML 2025.
>
>[2] Initialization is Critical to Whether Transformers Fit Composite Functions by Reasoning or Memorizing. NeurIPS 2024.


>w2: The description of SFT in Lines 116-136 is overly detailed and tangential to the paper's contributions. This section could be significantly condensed to focus the reader on the novel aspects of this work.

>w3: Line 155-156 "Modules with fewer trainable parameters (e.g., attention-only) are trained for more iterations" Why use matched FLOPs rather than matched parameters with fixed iterations?

>w4: The experimental design bears substantial similarity to that of [3]. The core setup, including the benchmarks (GP, V-IRL) and the evaluation methodology (ID/OOD split), closely follows prior work without significant innovation.
>
>[3] Sft memorizes, rl generalizes: A comparative study of foundation model post-training.

>w5: **Please standardize the citation style**. The manually formatted citation on Line 202 is an example. All in-text citations must use the required command (e.g., \citep \citet) to ensure they correctly link to the reference list.

>w6: Why "restrict fine-tuning to the first L transformer blocks" (Line 286) rather than others?

>w7: Line 350-352 "our experiments demonstrate that substantially smaller learning rates can alter the OOD trend, allowing SFT to match or even surpass reinforcement learning in OOD settings. " The claim that SFT outperforms RL **lacks evidentiary support**, as the experimental setup omits critical details of the RL baseline. Key implementation specifics (e.g., algorithm type - PPO or DPO, reward function design, hyperparameters) are not provided. To make this comparison credible and avoid overclaiming, the authors must first detail their RL methodology. Furthermore, a controlled comparison under an identical learning rate schedule is essential to isolate the true effect of the fine-tuning method, separate from hyperparameter tuning.

**Questions:**

See Weaknesses.

---

### Official Review · Reviewer_MgiJ · 2025-10-30

**Soundness:** 1
**Presentation:** 1
**Contribution:** 2
**Rating:** 0
**Confidence:** 3

**Summary:**

The authors provide an empirical study on which modules of a Transformer contribute to memorization or generalization in SFT. Selective finetuning of modules of the transformer architecture has been done to analyze the OOD generalization.

**Strengths:**

* Provides an analysis of which module of the transformer architecture is responsible for memorization and generalization
* Provides insights on how to avoid overfitting during SFT

**Weaknesses:**

* Writing could be significantly enhanced - the space has been used lavishly with so many bullet points. Table 1 confuses the reader and maybe even unnecessary, describing it in text format could have been clearer and would not occupy a lot of paper space
* Only two reasoning tasks have been considered, it is unclear if it will generalize to other LLM tasks
* The attention-only modules have been trained for more iterations to match the FLOPs of FNN or full model. The improved performance of attention-only fine-tuning may be due to longer training rather than because of the module choice

**Questions:**

* Under matched number of trainable parameters, were the models trained for the same number of iterations?
* Whether the choice of layers to finetune in the parameter-matched setup influence the results? If so, how to choose the layers to finetune?

---

### Official Review · Reviewer_n3RY · 2025-11-02

**Soundness:** 2
**Presentation:** 3
**Contribution:** 2
**Rating:** 4
**Confidence:** 4

**Summary:**

This paper studies and analyzes out-of-distribution (OOD) generalizations after SFT. Specifically, it investigates how different Transformer modules contribute to memorization and generalization during SFT. Through controlled experiments on two reasoning benchmarks (GeneralPoints and V-IRL), the authors show that fine-tuning only the attention layers preserves OOD generalization. In addition, they also find that using smaller learning rates mitigates memorization collapse, enabling SFT to achieve better OOD performance.

**Strengths:**

1. The paper provides a clear and systematic analysis of how different Transformer modules (attention vs. feedforward) affect memorization and OOD generalization during supervised fine-tuning.
2. The experimental design is well-controlled, using both arithmetic and navigation reasoning tasks with matched compute budgets to ensure fair comparisons.
3. The findings are practical, showing that attention-only fine-tuning and smaller learning rates can effectively mitigate overfitting and improve generalization.

**Weaknesses:**

1. While the empirical findings are insightful and provide a clearer understanding of how different Transformer modules behave during SFT, the overall contribution remains incremental. The paper does not introduce a novel algorithm or theoretical framework, and the conclusions (e.g., attention layers support generalization, smaller rates benefit OOD performance) are relatively intuitive.
2. It seems that all experiments focus on the SFT effect of a VLM (Llama-3.2-Vision-11B). More models or using an LLM on the pure language benchmarks (e.g., GP-L, V-IRL-L) can provide more in-depth insights on SFT.

**Questions:**

Is there any theoritical explanation about **why** fine-tuning only attention layers preserves OOD performance?

---

### Official Review · Reviewer_9ci7 · 2025-11-03

**Soundness:** 1
**Presentation:** 2
**Contribution:** 1
**Rating:** 0
**Confidence:** 4

**Summary:**

This paper investigates why supervised fine-tuning (SFT) of large language models often leads to overfitting and poor out-of-distribution (OOD) generalization. Through controlled experiments on reasoning benchmarks (GeneralPoints and V-IRL), it analyzes how different Transformer modules—attention and feedforward networks (FNNs)—contribute to memorization. The authors find that fine-tuning only the attention layers substantially improves OOD generalization, while FNN-only or full-parameter SFT collapses due to memorization. Even under matched training FLOPs and parameter budgets, attention-only SFT maintains higher OOD accuracy. Additionally, smaller learning rates mitigate memorization and allow SFT to achieve or exceed reinforcement learning–based alignment methods. The study concludes that selective, low-rate attention fine-tuning is a simple yet effective way to enhance generalization.

**Strengths:**

* Clear empirical focus on module-level dynamics in SFT.
* Well-controlled experiments (matched FLOPs, parameter counts).
* Practical implications for efficient alignment without RL.

**Weaknesses:**

Experiments are limited to two tasks (GP, V-IRL) with narrow reasoning domains; generality across language understanding or open-ended tasks is unclear.

The claim that attention-only tuning causes better OOD generalization is correlational; causal mechanisms (e.g., gradient flow or representation analysis) are not deeply examined.

No ablation on specific attention sub-components (Q/K/V projections, output linear layers).

Lack of statistical significance reporting or variance across runs.

The comparison to RL methods (e.g., PPO/DPO) is shallow—does not equalize reward signal or data diversity.

There have lots of PEFT works on selective FT that should be compared with. Some results in these previous work also show the weakness of FULL SFT.

**Questions:**

Would combining attention-only SFT with lightweight FNN adaptation (e.g., LoRA-FNN) yield a balance between memorization and flexibility?

How would the observed dynamics change on larger, less synthetic datasets (e.g., reasoning over natural text or multimodal benchmarks)?

---

### Note · Authors · 2025-11-30

I have read and agree with the venue's withdrawal policy on behalf of myself and my co-authors.